# H(N3)dap (Hdap = 2,6-Diaminopurine) Recognition by Cu_2_(EGTA): Structure, Physical Properties, and Density Functional Theory Calculations of [Cu_4_(μ-EGTA)_2_(μ-H(N3)dap)_2_(H_2_O)_2_]·7H_2_O

**DOI:** 10.3390/molecules28176263

**Published:** 2023-08-26

**Authors:** Homa Mousavi, María Eugenia García-Rubiño, Duane Choquesillo-Lazarte, Alfonso Castiñeiras, Luis Lezama, Antonio Frontera, Juan Niclós-Gutiérrez

**Affiliations:** 1Department of Inorganic Chemistry, Faculty of Pharmacy, University of Granada, 18071 Granada, Spain; y5245969m@correo.ugr.es; 2Departamento Fisicoquímica, Facultad de Farmacia, Universidad de Granada, 18071 Granada, Spain; rubino@ugr.es; 3Laboratorio de Estudios Cristalográficos, IACT, CSIC-Universidad de Granada, Avda. de las Palmeras 4, 18100 Armilla, Spain; duane.choquesillo@csic.es; 4Department of Inorganic Chemistry, Faculty of Pharmacy, University of Santiago de Compostela, 15782 Santiago de Compostela, Spain; alfonso.castineiras@usc.es; 5Departamento de Química Orgánica e Inorgánica, Facultad de Ciencia y Tecnología, Universidad del País Vasco, UPV/EHU, 48940 Leioa, Spain; luis.lezama@ehu.es; 6Departament de Química, Universitat de les Illes Balears, Crta. de Valldemossa km 7.5, 07122 Palma de Mallorca, Spain; toni.frontera@uib.es

**Keywords:** diaminopurine, molecular recognition, DFT calculations, Cu complexes

## Abstract

Reactions in water between the Cu_2_(µ-EGTA) chelate (EGTA = ethylene-bis(oxyethyleneimino)tetraacetate(4-) ion) and Hdap in molar ratios 1:1 and 1:2 yield only blue crystals of the ternary compound [Cu_4_(μ-EGTA)_2_(μ-H(N_3_)dap)_2_(H_2_O)_2_]·7H_2_O (**1**), which has been studied via single-crystal X-ray diffraction and various physical methods (thermal stability, spectral and magnetic properties), as well as DFT theoretical calculations. In the crystal, uncoordinated water is disordered. The tetranuclear complex molecule also has some irrelevant disorder in an EGTA-ethylene moiety. In the complex molecule, both bridging organic molecules act as binucleating ligands. There are two distorted five- and two six-coordinated Cu(II) centers. Each half of EGTA acts as a tripodal tetradentate Cu(II) chelator, with a mer-NO2 + O(ether, distal) conformation. Hdap exhibits the tautomer H(N3)dap, with the dissociable H-atom on its less basic N-heterocyclic atom. These features favor the efficient cooperation between Cu-N7 or Cu-N9 bonds with appropriate O-EGTA atoms, as N6-H···O or N3-H···O interligand interactions, respectively. The bridging role of both organics determines the tetranuclear dimensionality of the complex. In this crystal, such molecules associate in zig-zag chains built by alternating π–π interactions between the five- or six-atom rings of Hdap ligands of adjacent molecules. DFT theoretical calculations (using two different theoretical models and characterized by the quantum theory of “atoms in molecules”) reveal the importance of these π–π interactions between Hdap ligands, as well as those corresponding to the referred hydrogen bonds in the contributed tetranuclear molecule.

## 1. Introduction

The consideration of EGTA as a chelator is based on the fact that each half can function as a tetradentate ligand, featuring an N-substituted iminodiacetic group that includes an additional oxygen ether donor. On the other hand, Hdap is presently recognized as a biological precursor to other purines found in nucleosides, nucleotides, and nucleic acids [1,2,3,4,5,6,7,8,9]. The schematic structures of the ligands EGTA (ethylene-bis(oxyethyleneimino)tetraacetate(4-) ion), and Hdap are shown in Figure 1. Notably, Hdap is acknowledged for promoting DNA lesion repair under prebiotic conditions [4,5,6]. Additionally, Hdap nucleosides exhibit broad-spectrum antiviral and antibacterial properties [7], and endeavors have been made towards their synthesis [8]. Furthermore, there are available results regarding synthetic nucleotides with Hdap as prodrugs [9,10,11].

Turning to the acid H_4_EGTA and its complexes with metal ions in the Cambridge Structural Database (CSD), it is observed that the crystalline structure of the acid itself is accessible [12]. Moreover, approximately 50 crystalline structures containing the tetravalent anion EGTA^4−^ or its partially protonated form HEGTA^3−^ as a ligand are also available [13,14,15,16,17,18,19,20,21,22]. A comprehensive analysis of these structures reveals three general compound types: (i) Complexes where EGTA primarily functions as a chelator for a metal center. These compounds often exhibit a high coordination index (ranging from 8 to 10). Examples include complexes of yttrium(III) and most trivalent lanthanides (Ln = La-Lu), except for praseodymium and promethium (both radioactive with short half-lives), as well as lutetium (illustrated with a Gd example [13]), calcium, strontium, barium, cadmium, hafnium, and zirconium. (ii) Complexes where EGTA serves as a dinucleating bridge between two copper(II) [16,17,18,19] or two nickel(II) centers. (iii) Heterometallic combinations involving sophisticated clusters [20,21,22].

Diaminopurine (Hdap) has undergone investigation in approximately 40 structures, encompassing its protonated, molecular, and anionic forms [23,24,25,26]. Both the molecular and anionic forms exhibit a range of structures influenced by two factors that are also characteristic of other purines. Without delving into specifics, these factors encompass the well-established basicity of adenine nitrogen atoms (6-aminopurine), following the sequence N9 > N1 > N7 > N3 >> -N(6)H2. Additionally, the potential for tautomerism prompts the transfer of the proton typically bonded to N9 to other N-heterocyclic donors. Furthermore, the steric considerations linked to the hexocyclic amino groups must not be disregarded. For Hdap, these factors pose significant challenges in terms of its coordination via N1.

With these factors in consideration, the objective of this study was to synthesize a compound in which EGTA serves as a Cu(II) dinucleating bridge, enabling the incorporation of a pair of Hdap ligands at its termini. Magnetic measurements and density functional theory (DFT) calculations have been performed. The energetic features of two different π–π stacking modes observed for the Hdap in the solid state have been analyzed in detail. It has been previously reported that hybrid organic–inorganic materials generate interesting supramolecular architectures, governed by the interplay between metal–ligand coordination bonds and noncovalent interactions [27,28,29,30,31].

## 2. Results and Discussion

### 2.1. The Synthesis of the Binary Precursor and the New Compound ***1***

The synthesis of the new compound followed the standard approach adopted by our research group. The chelating compound is consistently available in its acidic form, suitable for various first-row transition metals (Mn-Zn) or post-transition metals (Cd, Pb), with varying degrees of success. For Cu(II), the azurite type (bluish) or malachite type (greenish) can be utilized, under the assumption that the Cu_2_CO_3_(OH)_2_ (molecular weight 221.1 g/mol) formula is practically applicable for determining the required amount of the copper(II) sources. This approach rests on the concept that the reaction between the copper hydroxycarbonate and the acidic form of the chelator (such as amino-polycarboxylic acids, amino acids, small peptides, etc.) will produce CO_2_ as a by-product that can be easily removed (achieved through stirring, heating the reaction mixture, and employing a controlled vacuum environment, usually conducted in an aqueous medium).

Two important points must be mentioned in relation to this work (as well as closely related studies). First, the use of bluish copper hydroxycarbonate may result in unreacted dark CuO, whereas malachite typically leaves behind unreacted green residue. Second, it is advisable to eliminate the minor amount of unreacted material by employing careful filtration before introducing additional coligands to the binary complex solution. This overarching strategy offers the advantage of minimizing undesired by-products (e.g., alkaline salts) while also enhancing the likelihood of successful crystallization. Furthermore, when the quantities of reagents employed do not precisely correspond to the stoichiometry of the final product based on molecular recognition principles, this discrepancy can be attributed to the excess free ligand that might be retained in the mother liquors.

Our experiment, documented in the experimental section of this work, illustrates a case study where an excessive amount of Hdap hinders the initial crystallization of the new compound **1**, resulting in a reduced yield.

### 2.2. Molecular and Supramolecular Structures in the Crystal

The structure determined through the crystallographic study, using a single crystal, faces the challenge of dealing with highly disordered lattice water molecules. Due to this disorder, these water molecules are not considered when establishing the molecular structure of the compound. Furthermore, as depicted in Figure 2a, one of the EGTA chelating ligands exhibits some disorder in an ethylene unit. While this aspect holds limited significance concerning the binding between the Cu_2_(EGTA) chelate and the Hdap nucleobase, it likely contributes to the observed non-equivalence of the four metal centers within the tetranuclear molecule (as shown in Figure 2b). The information provided in Table 1 indicates a relatively high level of quality in terms of crystallography.

Table 2 presents the bond distance data and relevant trans angles within the coordination environments. In summary, each µ-EGTA bridging chelator creates an elongated octahedral environment of the 4 + 2 type, as well as a penta-coordinate environment of the 4 + 1 type. To elaborate, EGTA-1 is associated with the Cu1 (4 + 1) and Cu4 (4 + 2) centers, while EGTA-2 corresponds to the Cu2 (4 + 1) and Cu3 (4 + 2) centers. Both the 4 + 1 and 4 + 2 environments incorporate the H(N3)dap N7 or N9 donors, adhering to conventional purine notation.

The tetranuclear nature of compound **1** fundamentally arises from the bridging mode of the H(N3)dap ligands. This implies that the formation of compound **1** is a consequence of the molecular recognition-based bridging mode of Hdap ligands, as opposed to producing an alternative compound where the Cu_2_(µ-EGTA) chelate coordinates an Hdap ligand to each metal center. The establishment of the tetranuclear molecule is facilitated by the tautomerization of the 2,6-diaminopurine proton, transitioning from its more basic donor (N9) to its less basic heterocyclic donor (N3). Hence, tautomerization plays an effective role in contributing to the molecular recognition mode observed in the investigated compound.

Table 3 presents the distance and angle data for the hydrogen bonds identified in this crystallographic study. Notably, it encompasses N^3^-H···O and N^6^-H···O-type interactions occurring between the Hdap ligands and the coordinated oxygen atoms of EGTA.

The crystallographic study further unveils the formation of 1D infinite assembly, which is illustrated in a simplified manner in Figure 3a, where two types of π–π stacking interactions are highlighted. In order to achieve a high level of structural resolution (SQUEEZE), the decision was made to prioritize structural accuracy over the precise localization of uncoordinated water molecules. Intriguingly, the π–π stacking interactions are structured such that two five-membered rings of the Hdap ligand interact with each other at one end, while a similar arrangement occurs with the six-membered rings at the other end. These π–π stacking interactions are depicted in the figure using dashed lines that connect the centroids of the involved rings and have been further analyzed below. Moreover, compound **1** also forms self-assembled dimers in the solid state dictated by H-bonding interactions, as depicted in Figure 3b. The coordinated water molecule (O27) interacts with two O-atoms of the coordinated carboxylate groups of EGTA (O22 and O65).

The coordination of the H(N3)dap tautomer has been previously documented in five compounds prior to the current study. The pertinent data have been compiled in Table 4, highlighting two key aspects: the molecular or polymeric composition of the compound and, where applicable, the occurrence of hydrogen bonds N3-H···O and N6-H···O that collaborate with M-N9 or M-N7 coordinate bonds (with M representing the transition metal).

The limited available data suggest the presence of analogous compounds involving copper, zinc, or cobalt. Among compounds of a molecular nature, data are solely obtainable from the MULCED compound [23], while the other compounds exhibit polymer structures (1D or 2D) or belong to the category of metal–organic frameworks (MOF) (3D). The existing data implies that the manner of molecular recognition demonstrated by compound **1** and its analogues is viable, irrespective of whether the known compounds exhibit a molecular or polymeric structure.

### 2.3. Physical Properties

The physical properties studied include thermal, spectral, and magnetic stability, which are elaborated on in the subsequent sections.

For comparison, Appendix A (Appendix A) illustrates the FT-IR spectra of both the binary compound [Cu_2_(µ-EGTA)(H_2_O)_2_]·2H_2_O (without Hdap) and compound **1**.

Notably, in the spectrum of compound **1** above 3000 cm^–1^, distinct absorptions can be identified due to antisymmetric and symmetric water stretching modes, along with the primary amino group. Additionally, two bands at 3194 and 3139 cm^–1^ correspond to N-H group stretching modes. Within this spectral region, the presence of water in the compound is evident, as well as the existence of two non-equivalent N-H groups corresponding to the N3-H group of the two slightly different tautomers of H(N3)dap found in the crystal. Furthermore, a broad band in the range of 1650 to 1620 cm^–1^ signifies the scissor-like deformation of both water molecules and the exocyclic amino groups of Hdap. This band also encompasses the antisymmetric stretching of the EGTA carboxylate groups.

Of particular diagnostic significance is a faint band at 1534 cm^–1^, indicating the in-plane deformation of the N3-H group, δ (N-H). This band emerges in a region with relatively few other absorptions. At 1372 cm^–1^, the symmetric stretching of the carboxylate groups is observed. Additionally, two noteworthy bands are identified at 1114 and 850 cm^–1^, corresponding to the antisymmetric and symmetric stretching vibrations of the C-O-C group. These bands typically appear in the range of 1300–1000 and 890–820 cm^–1^. In the case of the binary compound, they are observed at 1088 and 856 cm^–1^. The out-of-plane deformation band of the aromatic CH groups in Hdap is manifested as a peak at 850 cm^–1^. This particular band consistently appears within the range of 900–860 cm^–1^, which is deemed normal, especially in hydrocarbons. Moving on to Figure 4, the TGA (in an air environment) and DSC (under N2 atmosphere) curves are depicted. As is often the case, the process of losing the seven uncoordinated water molecules and the two distal aqua ligands occurs in a combined manner, corresponding to steps 1 and 2, as outlined in Table 5.

These processes occur through three thermal decomposition stages, resulting in the generation of carbon dioxide, carbon monoxide, water, and the three oxides of nitrogen commonly observed (N_2_O, NO, NO_2_), along with trace amounts of methane around 520 °C. At this temperature, an oxide of copper is produced, and its experimental value slightly surpasses the calculated value by less than 1%. The decomposition of N_2_O in the atmosphere distinctly illustrates the heat absorption dynamics, primarily corresponding to the coordinated or uncoordinated water loss between 30 °C and 180 °C. This observation indicates that the concurrent loss of both coordinated and uncoordinated water constitutes a complex and overarching process. Beyond 180 °C, a reasonably balanced interaction between heat absorption and heat loss processes emerges, coinciding with the thermal decomposition of the organic ligands. It is worth noting that the complete loss of water initiates thermal decomposition at around 200 °C, ultimately leading to the formation of copper oxide at 520 °C.

In terms of the electronic spectrum obtained through diffuse reflectance, an asymmetric d-d band is evident, displaying maximum absorption at 685 nm (refer to Figure 5). The barycenter of intensity for this band is situated at 735 nm, a value lower than the peak absorption of the corresponding electronic spectrum of the hexaaqua-cation (approximately 800 nm) [34]. In summary, this comparison highlights that the metal environment within compound **1** exhibits a more pronounced ligand field effect compared to the aqua-cation.

Electronic spin resonance (ESR) measurements were conducted in the temperature range of 5–300 K, utilizing both X and Q band frequencies. While the X-band ESR spectra display near-axial symmetry, a substantial level of rhombicity becomes apparent when operating at the Q-band frequencies (see Figure 6 and Figure 7). Employing a computer program that operates within the second order of perturbation theory, the spectra could be accurately fitted with a single g tensor. This outcome suggests that the exchange interaction between copper ions is of a significant magnitude, effectively collapsing the spectra of magnetically non-equivalent sites [35]. Notably, the intensity of the ESR signal experiences a significant decrease below 20 K, indicating that the dominant exchange interactions are primarily antiferromagnetic in nature. The primary components of the calculated g tensor are as follows: g_1_ = 2.235, g_2_ = 2.071, and g_3_ = 2.052 (g_II_ = 2.235; g_⊥_ = 2.061). The G parameter, defined by G = (g_II_ − 2)/(g_⊥_ − 2), is calculated as 3.8, which is less than 4. This value confirms the existence of exchange interactions and suggests a minor misalignment between the principal axes of individual g tensors within compound **1** [36]. Nevertheless, the lowest g value indicates a d_x2-y2_ ground electronic state for all the Cu(II) centers, which corresponds to the axially elongated 4 + 1 and 4 + 2 coordination environments.

The thermal evolution of the magnetic molar susceptibility is depicted in Figure 8. As the temperature decreases, the susceptibility progressively increases, reaching a maximum value of 0.0369 cm^3^/mol at 23 K. Subsequently, it experiences a rapid decline, reaching 0.003 cm^3^/mol at 5 K. The product χ_m_T remains nearly constant (1.56 cm^3^ K/mol) within the temperature range of 300 to 100 K, and then it rapidly diminishes upon cooling to 5 K (0.016 cm^3^ K/mol). The high-temperature data (T > 50 K) is well described by a Curie–Weiss law (χm = Cm/T-θ), with Cm = 1.62 cm^3^ K/mol and θ = −9.7 K (Figure 8, right). The observed Curie constant (Cm) aligns with calculations for four non-interacting S = 1/2 ions, utilizing the g value derived from the ESR study (g = 2.12). The negative θ value and the overall shape of the χmT vs. T curve indicate the prevalence of short-range antiferromagnetic interactions among the Cu(II) ions, leading to an S = 0 spin ground state.

Given the tetranuclear structure of compound **1**, various magnetic exchange pathways involving the H(N3)dap and EGTA organic ligands need to be considered: Cu1···Cu2 (5.783 Å), Cu3···Cu4 (5.676 Å), Cu1···Cu4 (7.453 Å), and Cu2···Cu3 (7.233 Å). As a result, the spin Hamiltonian presented below should be utilized to analyze the magnetic properties, encompassing solely isotropic interactions and the Zeeman term:(1)$$\hat{H}=-J_{12}(S_{1}.S_{2})-J_{34}(S_{3}.S_{4})-J_{14}(S_{1}.S_{4})-J_{23}(S_{2}.S_{3})+\mu_{\beta }\sum_{i=1}^{4}B.g_{i}.S_{i}$$

However, due to the substantial number of adjustable parameters, employing the above expression to fit the magnetic curves does not yield dependable values for the exchange coupling constants. As a result, introducing certain constraints on the Jij and g parameters becomes necessary. In this context, it has been postulated that despite minor disparities in the superexchange pathway, the two interactions propagated by the EGTA ligand are analogous (J_12_ = J_34_ = J_A_), and the same holds true for the two interactions mediated by the H(N3)dap ligand (J_14_ = J_23_ = J_B_). Furthermore, considering the findings from the ESR results, the assumption has been made that all local g-values are equivalent. Under these defined conditions, fitting the experimental susceptibility data using the full-matrix diagonalization PHI program [37] has yielded the ensuing set of parameters: J_A_ = –12.4 cm^–1^, J_B_ = –0.33 cm^–1^, and g = 2.102 with an R factor of 1.3 × 10^–4^. The calculated curve aligns remarkably well with the experimental data across the entire temperature range examined (Figure 8, left). The smaller coupling constant can be unequivocally attributed to interactions propagated through the EGTA ligand, as the magnetic orbitals (mainly d_x2−y2_) are situated in planes parallel to each other and perpendicular to the exchange pathway. The orbital overlap between the unpaired electron-bearing orbitals is notably more favorable through the H(N3)dap ligands within this compound, thus explaining the stronger antiferromagnetic couplings.

A recent work has analyzed the magnetic properties of two Mn(II) polymers with tetracarboxylate linkers (5,5′-(ethene-1,2-diyl)diisophthalic acid), also exhibiting antiferromagnetic coupling [38] with smaller J values (~–1.5 cm^–1^).

### 2.4. DFT Calculations

In the solid state, compound **1** forms infinite 1D assemblies that propagate via two different π–π stacking interactions (see Figure 9a). One is established between the 2,6-diaminopurine bases that are coordinated to the pseudo-octahedral Cu-atoms (denoted as A···A in Figure 9a) and the other binding mode involves the 2,6-diaminopurine rings that are coordinated to the square-pyramidal Cu-atoms (denoted as B···B). The theoretical study is devoted to the energetic analysis of both π–π stacking modes and the characterization of some additional H-bonds that are formed in the A···A binding mode. In order to keep the model computationally approachable and to simplify the description and DFT analysis of both interaction modes, the tetranuclear complex has been divided into two halves by cutting the μ-EGTA chelate as indicated in Figure 9b, resulting in two dinuclear complexes denoted as “**A**” and “**B**” (see Figure 9b).

First, the MEP surfaces of fragments **A** and **B** have been computed to explore the most electron rich and poor parts of the molecules, which are represented in Figure 10. As expected, the minima are located at the noncoordinated atoms of the carboxylate groups (–75 and –63 kcal/mol in **A** and **B**, respectively) and the MEP maxima are located at the amino group bonded to C2 (ranging +54 to +60 kcal/mol). The MEP value at the H-atoms of the Cu-coordinated water molecules is similar (+53 kcal/mol), revealing that both the NH_2_ groups and water molecules are strong H-bond donors. The MEP value at the available NH bond of the NH_2_ group at C6 is also large and positive (+46 and +44 kcal/mol in **A** and **B**, respectively) but significantly smaller than those at the NH_2_ groups due to the formation of the intramolecular NH···O H-bond in the former. The MEP is also positive at the –CH_2_– groups of the chelate ligand (+26 kcal/mol) and over the six-membered ring of the 2,6-diaminopurine.

The coordination via N7 and N9 in combination with the N3-H tautomer instead of the coordination via the more basic N1 or N3 in combination with the N9-H tautomer is likely due to the formation of extra H-bonds that compensate for the different coordination ability of the six- and five-membered ring N-atoms. This has been confirmed by performing QTAIM analysis. The distributions of bond critical points (CPs, fuchsia spheres) and bond paths of the Cu(II) complexes **A** and **B** are shown in Figure 11, confirming the existence of the NH···O contacts (via N3H3 and N6H6). The interaction energies of these NH···O HBs derived from the potential energy density (Vr) measured at the bond CPs range from –6.06 kcal/mol to –9.07 kcal/mol, confirming the energetic relevance of these contacts. Interestingly, the QTAIM analysis also evidences the presence of a weaker CH···O contact in the Cu(II)-atom coordinated by N7 in both models, where a bond CP and bond path connects the C8–H bond of the imidazole ring to the Cu-coordinated O-atom of the carboxylate group. The energy associated with these contacts are weaker (–5.08 and –3.70 kcal/mol in **A** and **B**, respectively).

As commented above, two different π–π stacking modes have been analyzed theoretically. They are shown in Figure 12 along with the QTAIM and NCIplot analyses. The combined QTAIM and NCIplot representations are very useful to visualize NCIs in real space, including their attractive/repulsive nature (provided by the color of the isosurfaces). It can be observed that the self-assembled A···A dimer shown in Figure 12a combines two strong and symmetrically equivalent OH···N interactions (established between the coordinated water molecules and the N1-atom of 2,6-diaminopurine) with an antiparallel π–π stacking interaction. The latter is characterized by four bond CPs and bond paths interconnecting C and N atoms and includes the exocyclic NH_2_ group. Moreover, a green (attractive) reduced density gradient (RDG) isosurface is located between the 2,6-diaminopurine moieties, further characterizing the π–π interaction. The dimerization energy is large and negative (–17.00 kcal/mol), confirming the importance of this assembly in the solid state of compound **2**. The contribution of the HBs is –14.4 kcal/mol, thus supporting the strong nature of these HBs, in line with the blue RDG isosurfaces located between the OH groups and N1-atoms. This energetic analysis reveals that in fact, the HBs dominate the formation of this dimer. The other π–π stacking binding mode (B····B) that involves the 2,6-diaminopurine coordinated to the square-pyramidal Cu(II) metal centre is much complicated. The RDG isosurface is very extended and occupies most of the space between the monomers, disclosing a strong complementarity. Moreover, there is an intricate combination of interactions taking place, explaining the very large dimerization energy (–44.5 kcal/mol), more than twice the A···A. This is due to the existence of multiple CH··O contacts between the carboxylate groups and the CH bonds of the chelator. Moreover, two ancillary anion–π interactions (indicated as O···π in Figure 12b) are also observed, characterized by a bond CP and bond path connecting one O-atom of the carboxylate group to the pyrimidine ring, in line with the MEP analysis that revealed the π-acidity of this ring. The large binding energy observed for the B···B dimer likely explains the pentacoordination of these Cu(II) in B since the formation of the B···B staking mode is able to compensate for the stabilization that would be obtained via the coordination of the water molecules.

## 3. Materials and Methods

### 3.1. Reagents and Synthesis for the Dicopper(II)-EGTA Chelate and the New Compound **1**

Both products have been synthesized through the reaction between Cu_2_CO_3_(OH)_2_ (green malachite, Aldrich, Darmstadt, Germany) and H_4_EGTA (Aldrich, Darmstadt, Germany), either in the absence or presence (for the synthesis of **1**) of 2,6-diaminopurine (Aldrich, Darmstadt, Germany).

The binary dicopper(II)-EGTA chelate [Cu_2_(µ-EGTA)(H_2_O)_2_]·2H_2_O (molecular weight 575.47 g/mol) has been successfully prepared in good yields using the previously reported strategy from our research group [16]. In a recent experiment, 0.44 g (2 mmol) of green malachite was reacted with H_4_EGTA (0.76 g, 2 mmol) in 125 mL of distilled water, within a stoppered Kitasato flask of 500 mL. The flask’s side outlet allowed the escape of the only by-product, CO_2_, preventing potential splattering. The mixture was continuously stirred and heated (45–50 °C) for approximately two hours until unreacted malachite was no longer observable. The stoichiometry of the reaction is as follows:Cu_2_CO_3_(OH)_2_ + H_4_EGTA + H_2_O → [Cu_2_(µ-EGTA)(H_2_O)_2_]·2H_2_O + CO_2_↑.

The blue solution is allowed to cool to room temperature and is then filtered (without using a vacuum) through a Büchner funnel into a 250 mL crystallization flask. Subsequently, it is covered with a perforated plastic film to regulate the solvent evaporation. After two months, the chelate is collected and air-dried, yielding 0.87 g (1.52 mmol, 75.6%).

Initially, a similar procedure was employed for the synthesis of the new compound **1**, with the exception of using the aforementioned quantities of malachite and H_4_EGTA in a larger volume of distilled water (175 mL). To eliminate trace amounts of unreacted malachite, the resulting solution of Cu_2_(µ-EGTA) chelate was filtered into an Erlenmeyer flask. Subsequently, Hadp (0.30 g, 2 mmol) was incrementally added to this solution at room temperature, followed by filtration into an appropriate crystallization flask. This approach signifies the use of Hadp in a 100% excess compared to the stoichiometry of compound **1**.
2 Cu_2_CO_3_(OH)_2_ + 2 H_4_EGTA + 2 Hdap + 15 H_2_O → 2 [Cu_4_(µ-EGTA)_2_(µ-Hdap)_2_(H_2_O)_2_]·7H_2_O + CO_2_↑

The excess of Hadp remained in the mother liquors, resulting in a reduced practical yield (approximately 50%). However, this excess enabled the exclusive production of high-quality compound **1** (Appendix A). In contrast, employing a ‘stoichiometric amount’ of Hadp (i.e., 1 mmol of Hadp, 0.15 g) initially yielded crystals of the binary chelate, as confirmed via single-crystal X-ray diffraction.

Elemental analysis (%): Calc. for C_38_H_70_Cu_4_N_16_O_29_, C 31.06, H 4.80, N 15.19. Found, C 31.02, H, 4.85, N, 15.19.

### 3.2. Physical Measurements

Elemental analysis was conducted using a Thermo Scientific Flash 2000 instrument (Thermo Fisher Scientific Inc., Waltham, MA, USA). Infrared spectra were recorded by placing samples in KBr pellets and employing a Jasco FT-IR 6300 spectrometer (Jasco Analítica, Madrid, Spain). Electronic spectra via diffuse reflectance were obtained using a Varian Cary-5E spectrophotometer (Agilent Scientific Instruments, Santa Clara, CA, USA) with ground crystalline samples. Thermogravimetric analyses (TGA) were performed under an air-dry flow (100 mL/min) with a Mettler-Toledo TGA/DSC1 thermobalance (Mettler-Toledo, Columbus, OH, USA), using a heating rate of 10 °C/min. A series of 30 time-spaced FT-IR spectra were concurrently recorded during the experiment to detect evolved gases, utilizing a coupled FT-IR Nicolet 550 spectrometer (Thermo Fisher Scientific Inc., Waltham, MA, USA). Differential scanning calorimetry (DSC) measurements were carried out using a DSC-SHIMADZU mod. DSC-50Q instrument (Shimadzu Europe, F.R. Germany GbmH) in an N_2_ atmosphere, with a temperature range of 30–400 °C and a heating rate of 10 °C/min.

For room-temperature X-band electron spin resonance (ESR) spectra of powdered samples, a Bruker ELEXSYS E500 spectrometer equipped with a super-high-Q resonator ER-4123-SHQ was utilized. The magnetic field was calibrated using an NMR probe, and the frequency within the cavity was determined using an integrated MW-frequency counter. Q-band measurements were conducted on a Bruker ESP300 spectrometer equipped with an ER-510-QT resonator, a Bruker BNM 200 gaussmeter, and a Hewlett-Packard 5352B microwave frequency counter. Data collection and processing were carried out using the Bruker Xepr suite.

Magnetic measurements of powdered samples were performed within a temperature range of 5–300 K using a Quantum Design MPMS-7 SQUID magnetometer with a magnetic field of 0.1 T. Diamagnetic corrections were computed based on Pascal tables.

### 3.3. Crystallography

A blue plate crystal of **1** was affixed onto a glass fiber and employed for data collection. Crystal data were gathered at 298(2) K using a Bruker D8 VENTURE diffractometer. Throughout the process, graphite monochromated CuKα radiation (λ = 1.54184 angstroms) was utilized. The obtained data underwent processing with APEX3 [39] and were subsequently corrected for absorption through the application of SADABS (transmission factors: 1.000–0.714) [40].

The structure was initially solved using direct methods employing the XT program [41], followed by refinement utilizing full-matrix least-squares techniques against F2 with the XL program [42]. Positional and anisotropic atomic displacement parameters were refined for all non-hydrogen atoms. Hydrogen atoms were located through difference maps and were included with fixed contributions riding on their corresponding attached atoms, with isotropic thermal parameters set at 1.2/1.5 times those of the carrying atoms. To address the density contribution of disordered solvent molecules, the SQUEEZE option was employed [43], revealing the presence of approximately seven water molecules per asymmetric unit. Subsequent refinement iterations led to improved R factors and parameter errors, with significant enhancements in addressing the solvent disorder. The criteria for a satisfactory and comprehensive analysis included ratios of root-mean-square shifts to standard deviations below 0.001 and the absence of notable features in the final difference maps.

Atomic scattering factors were sourced from “International Tables for Crystallography” [44], and molecular graphics were generated using PLATON [45]. A summary of the crystal data, experimental particulars, and refinement outcomes are provided in Table 1.

### 3.4. Computational Details

Non-covalent interaction calculations were conducted utilizing Gaussian-16 [46] at the PBE0-D3/def2-TZVP level [47,48,49]. Solid-state interactions were evaluated using crystallographic coordinates. Binding energies were determined by comparing isolated monomer energies with their assemblies, while the Boys–Bernardi method [50] corrected for basis set superposition error (BSSE). Bader’s “Atoms in molecules” theory (QTAIM) [51], through the AIMAll calculation package [52], explored π-hole and H-bonding interactions. Molecular electrostatic potential surfaces (isosurface 0.001 a.u.) were generated using Gaussian-16 [47].

To assess the nature of interactions in terms of attraction or repulsion and visualize them in real space, the NCIPLOT index was employed [53]. This method, based on the NCI (Non-Covalent Interactions) visualization index derived from electronic density, utilizes the reduced density gradient (RDG) derived from density and its first derivative. This RDG is plotted against the density (represented as isosurfaces) across the molecule of interest.

Identifying attractive/stabilizing (blue-green isosurfaces) or repulsive (yellow-red isosurfaces) interactions is achieved using 3D plots through the sign of the second Hessian eigenvalue multiplied by the electron density (sign(λ_2_)ρ in atomic units). The specific NCIplot index parameters are as follows: RDG = 0.5; ρ cutoff = 0.04 a.u.; color range: –0.04 a.u. ≤ sign(λ_2_)ρ ≤ 0.04 a.u.

The level of theory and methodology used in this work has been previously used to analyze similar interactions in the solid state [54,55].

## 4. Concluding Remarks

In this manuscript, we present the synthesis and X-ray characterization of a new tetranuclear Cu(II) complex molecule, featuring two μ-EGTA chelators as binding ligands and two molecules of 2,6-diaminopurine in the tautomer H(N3)dap form. In the crystal, the presence of non-coordinated water introduces disorder. Nonetheless, our findings provide a foundation for discussing both molecular recognition, which underscores the rationale behind the complex molecule’s tetranuclear nature, and supramolecular recognition, elucidating the formation of chains through π–π interactions between pairs of 5- or 6-membered rings of H(N3)dap.

We systematically analyze the spectral, magnetic, and thermal stability properties of the compound based on the tetranuclear molecule’s structural characteristics. Additionally, a theoretical DFT study is conducted, shedding light on the significance of π–π stacking and H-bonding interactions in the solid state. This analysis employs two representative models. Notably, we explore O···π interactions and additional CH···O interactions in the B···B binding mode, which offer insights into the substantial dimerization energy observed. These interactions also account for the distinct coordination of the Cu(II) atoms, excluding the distal water molecules. This unique coordination pattern facilitates the approximation of 2,6-diaminopurine units, leading to a more effective monomer approximation.

Furthermore, we employ the QTAIM analysis to quantify intramolecular H-bonds in both model complexes. This quantification aids in rationalizing the complexation through N7 and N9 to the metal centers.

## Figures and Tables

**Figure 1 molecules-28-06263-f001:**
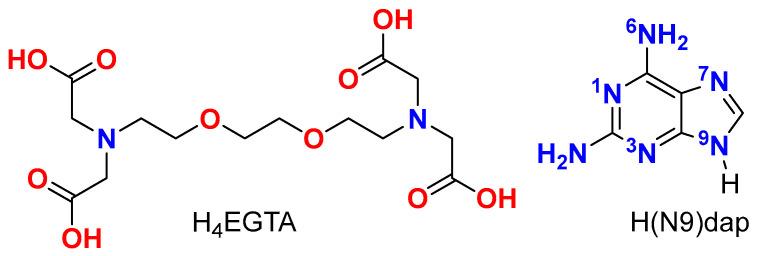
Schematics of H_4_EGTA (**left**) and H(N9)dap (**right**).

**Figure 2 molecules-28-06263-f002:**
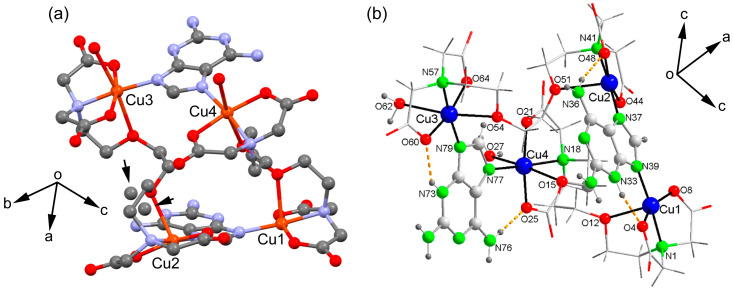
(**a**) EGTA chelating ligands with disorder in an ethylene unit. (**b**) The tetranuclear molecule [Cu_4_(μ-EGTA)_2_(μ-H(N3)dap)_2_(H_2_O)_2_] with its four metal centers is represented including the atom numbering scheme.

**Figure 3 molecules-28-06263-f003:**
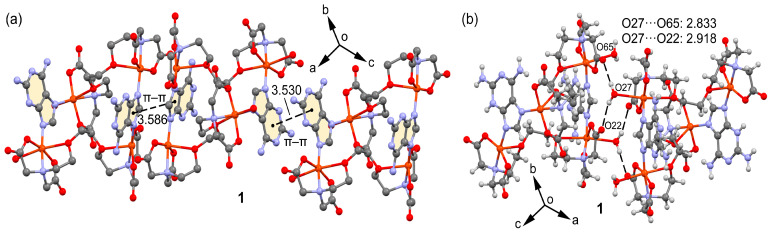
Partial views of the X-ray structure of 1 showing the formation of infinite 1D assemblies (**a**) and H-bonded self-assembled dimers (**b**). Distances in Å.

**Figure 4 molecules-28-06263-f004:**
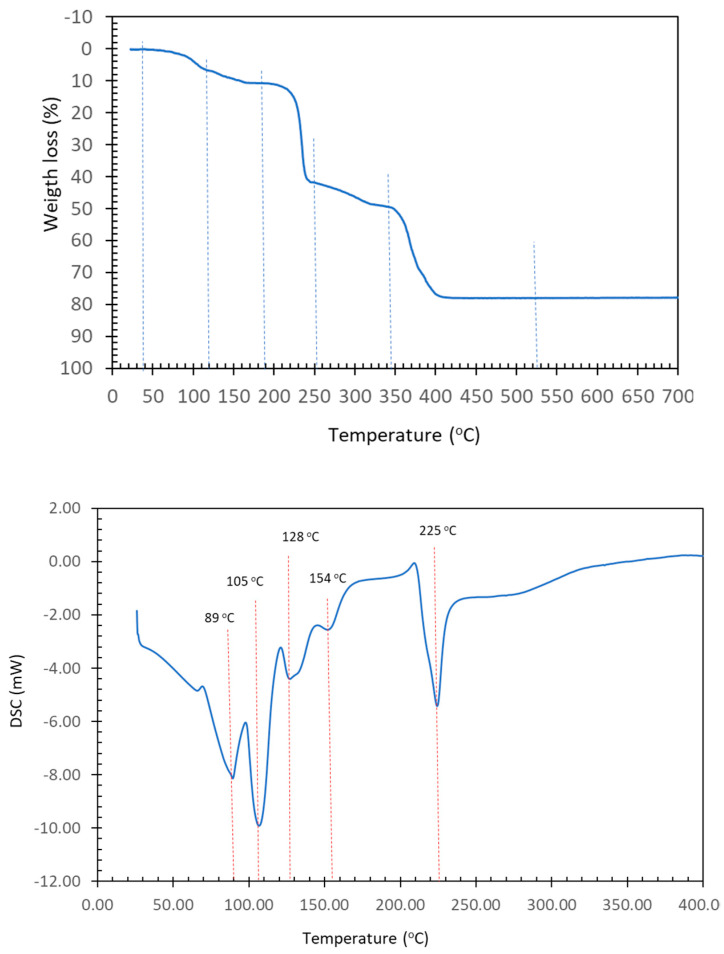
TGA (**up**) and DSC (**down**) curves of compound (**1**).

**Figure 5 molecules-28-06263-f005:**
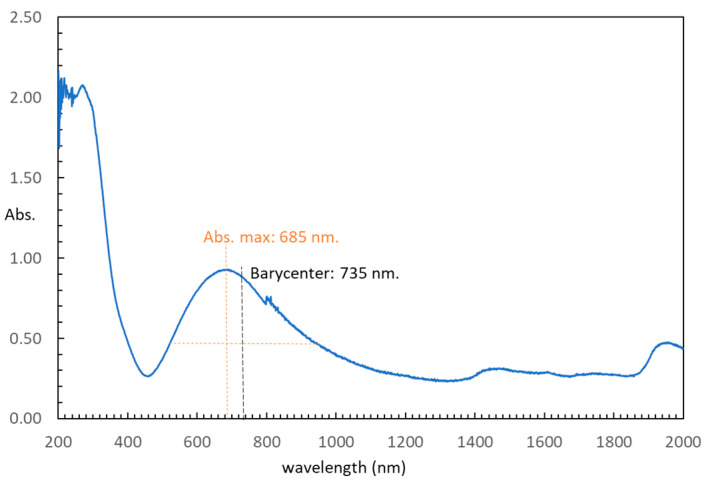
Electronic spectrum (diffuse reflectance) of compound (**1**).

**Figure 6 molecules-28-06263-f006:**
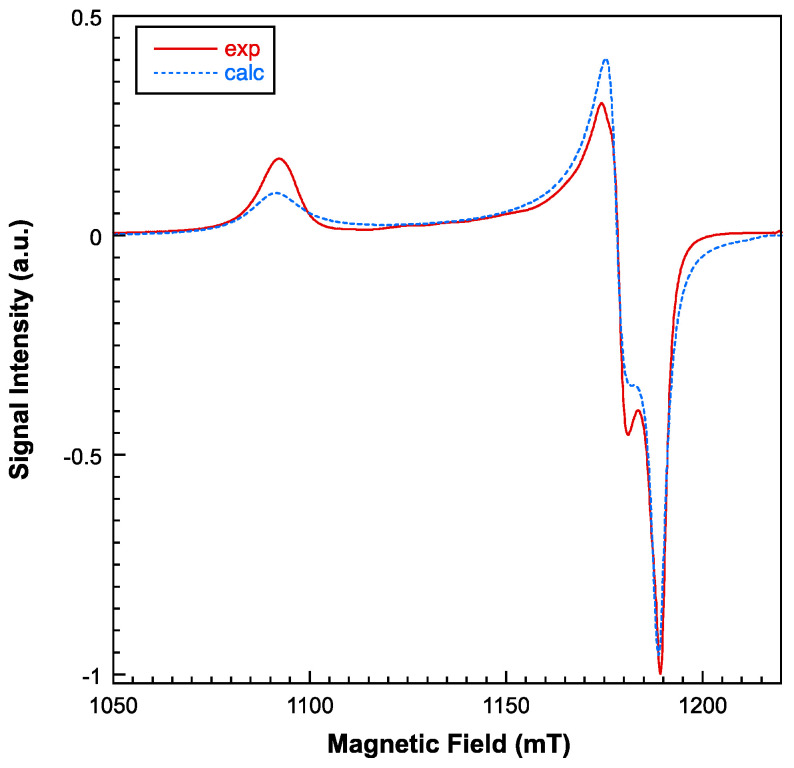
Experimental (continuous line) and simulated (dashed line) Q-band room-temperature ESR spectrum for compound **1**.

**Figure 7 molecules-28-06263-f007:**
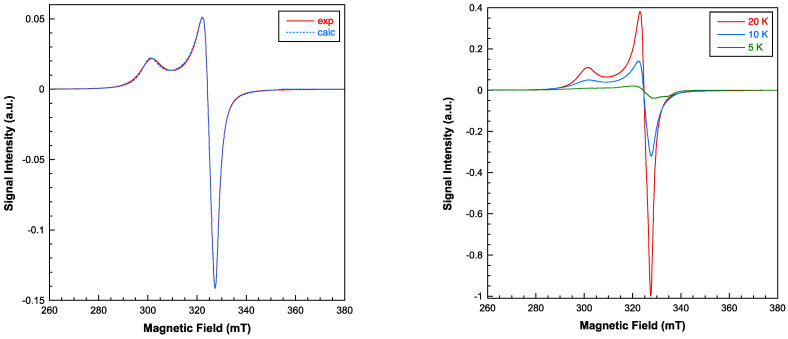
Experimental (continuous line) and simulated (dashed line) X-band room-temperature EPR spectrum for compound **1** (**left**). Thermal evolution of the signal at low temperatures (**right**).

**Figure 8 molecules-28-06263-f008:**
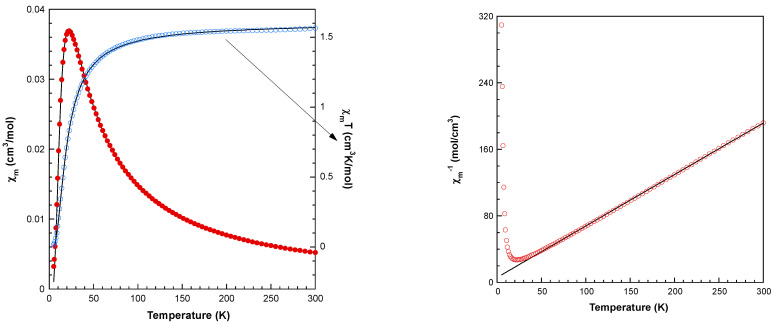
(**Left**): Magnetic behavior of compound **1** (see text for the details of fit). χ_m_ values represented as blue circles and the fitting as a solid line. The χ_m_T as red circles and the fitting as solid line. (**Right**): Thermal dependence of the reciprocal molar susceptibility for compound **1**. Continuous line corresponds to the Curie–Weiss fit for T > 50 K.

**Figure 9 molecules-28-06263-f009:**
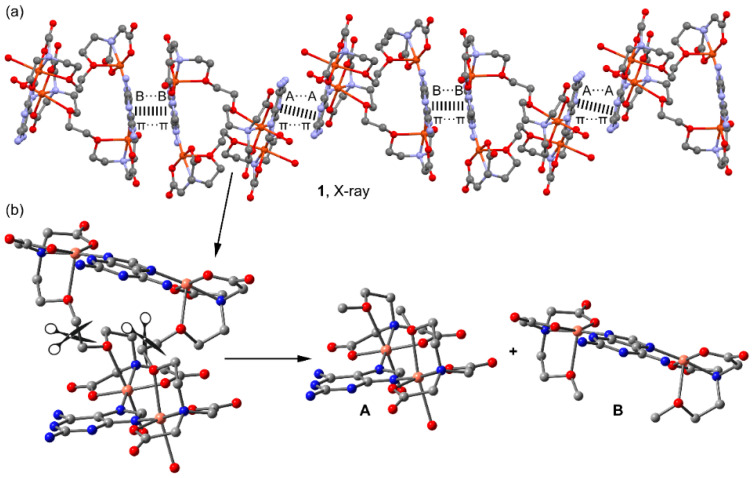
(**a**). Partial view of the X-ray structure of compound **1** where the 1D supramolecular chain is represented. (**b**) Dinuclear models A and B generated from the tetranuclear macrocycle.

**Figure 10 molecules-28-06263-f010:**
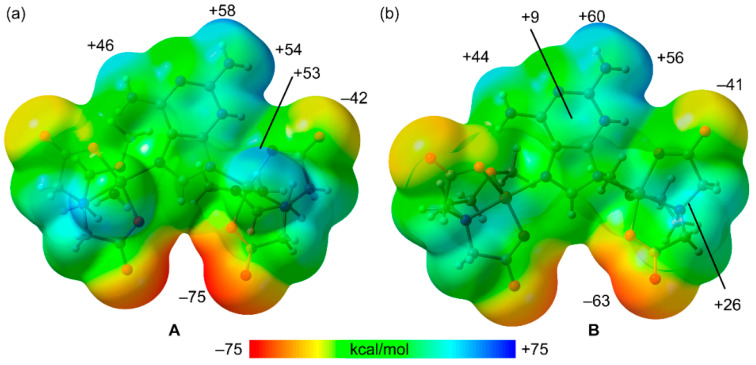
MEP surfaces of dinuclear model systems **A** (**a**) and **B** (**b**) at the PBE0-D3/def2-TZVP level of theory (density isovalue is 0.001 a.u.). The energies are given in kcal/mol.

**Figure 11 molecules-28-06263-f011:**
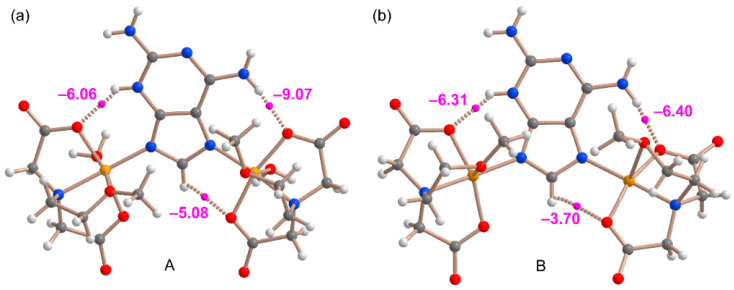
QTAIM analysis of the complexes **A** (**a**) and **B** (**b**) where the bond critical points (CPs, fuchsia spheres) and bond paths (dashed bonds) of the noncovalent interactions are shown. The energy of the HBs is indicated using a fuchsia colored font adjacent to the bond CPs.

**Figure 12 molecules-28-06263-f012:**
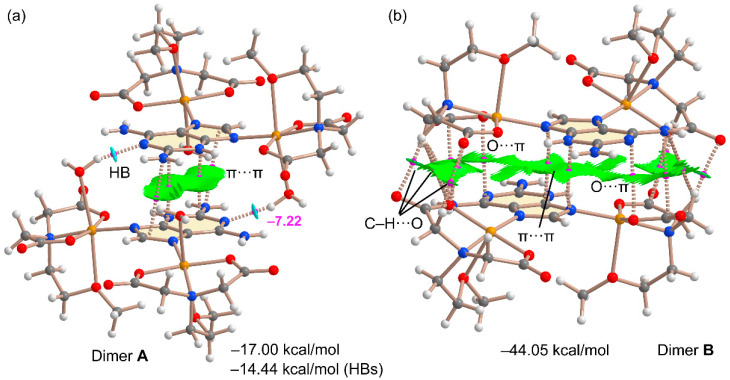
QTAIM/NCIPlot analysis of intermolecular bond CPs (fuchsia spheres), bond paths and RDG isosurfaces of the A···A (**a**) and B···B (**b**) π–π stacking binding modes of compound **2**. The individual interaction energies of the H-bonds are indicated using a red font next to the bond CPs.

**Table 1 molecules-28-06263-t001:** Crystal data and structure refinement for compound (**1**).

Empirical formula	C_38_H_70_Cu_4_N_16_O_29_	
Formula weight	1469.24	
Temperature	298(2) K	
Wavelength	1.54178 Å	
Crystal system	Triclinic	
Space group	*P* $\bar{1}$	
Unit cell dimensions	*a* = 12.9272(4) Å	*α* = 104.372(1)°
	*b* = 13.1727(4) Å	*β* = 94.015(2)°
	*c* = 19.6298(6) Å	*γ* = 95.213(2)°
Volume	3209.80(17) Å^3^	
Z	2	
Calculated density	1.520 Mg/m^3^	
Absorption coefficient	2.396 mm^–1^	
F(000)	1516	
Crystal size	0.10 × 0.10 × 0.09 mm	
θ range for data collection (°)	2.334 to 66.591.	
Limiting indices	14 ≤ h ≤ 15,−15 ≤ k ≤ 15,−23 ≤ l ≤ 23	
Reflections collected/unique	23,305/10,915 [*R*_int_ = 0.0287]	
Completeness to theta = 66.591°	96.2%	
Absorption correction	Semi-empirical from equivalents	
Max. and min. transmission	1.000 and 0.714	
Refinement method	Full-matrix least-squares on *F*^2^	
Data/restraints/parameters	10,915/1/742	
Goodness-of-fit on F^2^	1.038	
Final R indices [I > 2σ(I)]	*R*_1_ = 0.0354,	*wR*_2_ = 0.0956
R indices (all data)	*R*_1_ = 0.0394	*wR*_2_ = 0.0986
Largest diff. peak and hole	0.462 and −0.429 e·Å^−3^	
CCSD code	2287840	

**Table 2 molecules-28-06263-t002:** Selected coordination bond lengths [Å] and trans-angles [°] for **1**.

Cu(1)-N(1)	2.000(2)	Cu(2)-N(37)	1.973(2)
Cu(1)-O(4)	1.934(2)	Cu(2)-N(41)	2.004(3)
Cu(1)-O(8)	1.930(2)	Cu(2)-O(44)	1.939(2)
Cu(1)-N(39)	1.961(2)	Cu(2)-O(48)	1.951(2)
Cu(1)-O(12)	2.337(2)	Cu(2)-O(51)	2.424(2)
N(39)-Cu(1)-N(1)	θ = 175.67(10)	N(37)-Cu(2)-N(41)	θ = 174.95(10)
O(8)-Cu(1)-O(4)	φ = 158.40(11)	O(44)-Cu(2)-O(48)	φ = 163.30(10)
τ = (θ − φ)/60 *	0.29	τ = (θ − φ)/60	0.19
Cu(3)-N(57)	1.997(2)	Cu(4)-N(77)	1.9801(2)
Cu(3)-O(60)	1.969(2)	Cu(4)-N(18)	2.010(2)
Cu(3)-O(64)	1.955(2)	Cu(4)-O(21)	1.959(2)
Cu(3)-N(79)	1.944(2)	Cu(4)-O(25)	1.959(2)
Cu(3)-O(54)	2.417(2)	Cu(4)-O(27)	2.499(2)
Cu(3)-O(62)	2.552(3)	Cu(4)-O(15)	2.561(2)
N(79)-Cu(3)-N(57)	172.71(8)	O(27)-Cu(4)-O(15)	172.58(6)
O(54)-Cu(3)-O(62)	171.82(8)	N(77)-Cu(4)-N(18)	170.55(8)
O(64)-Cu(3)-O(60)	167.33(7)	O(21)-Cu(4)-O(25)	166.11(6)

* Addison–Reedijk parameter [32,33].

**Table 3 molecules-28-06263-t003:** Distances (D···A, Å) and angles (<, °) for hydrogen bindings in the crystal of **1**.

D-H···A	d(D···A)	<(D-H···A)
O(27)-H(27A)···O(65)#1	2.833(3)	159.4
O(27)-H(27B)···O(22)#1	2.917(3)	142.7
N(33)-H(33)···O(4)	2.760(3)	138.6
N(36)-H(36A)···O(26)#2	2.938(3)	159.4
N(36)-H(36B)···O(48)	2.844(4)	160.3
O(62)-H(62B)···N(71)#3	2.947(3)	162.0
N(72)-H(72A)···O(22)#4	2.751(3)	133.2
N(73)-H(73)···O(60)	2.781(2)	140.3
N(76)-H(76B)···O(25)	2.825(3)	170.4

Symmetry transformations to generate equivalent atoms: #1 = −x + 1, −y + 1, −z; #2 = x, y + 1, z; #3 = −x, −y + 1, −z; #4 = x − 1, y, z.

**Table 4 molecules-28-06263-t004:** Relevant H-bonding data for mixed-ligand metal complexes having as μ_2_-N7,N9 the tautomer H(N3)dap.

Compound ^§^	Formula *	N3-H···O (Å, °)	N6-H···O (Å, °)	Ref.
1 (this work)	[Cu_4_(μ-EGTA)_2_(μ-H(N3)dap)_2_(H_2_O)_2_]·7H_2_O	2.760(3), 139 2.781(2), 140	2.844(4), 139 2.825(3), 170.4	-
QUDKEG	[Zn_2_(μ_2_-Hdap)(tp)_2_]_n_ (3D MOF)	2.739(2), 173	3.003(2), 162 2.912(2), 155	[22]
QUDKIK	{[Zn_2_(μ_2_-Hdap)(tm)(μ_2_-OH)]·3H_2_O}_n_ (1D polymer)	2.559(2), 176	2.935(3), 172	[22]
MULCED	[Cu_2_(BCBC)_2_(μ_2_-N7,N9)Hdap)(H_2_O)_2_]·4H_2_O	2.732(7), 140	2.774(7), 169	[23]
FINDAC	[Zn(FDC)(μ_2_-N7,N9)Hdap)]·0.5H_2_O (3D MOF)	N/A	N/A	[24]
KOZNAR	[Co(Hdap)(ip)]_n_ (2D layers)	N/A	N/A	[25]

^§^ Acronyms in CSD for previously reported compounds. * Ligands: Hdap. tp = terephthalate(2-) ion. tm = trimesicate(3-) ion. BCBC = N,N-bis(carboxymethyl)-S-benzyl-cysteaminate(2-) anion. FDC = H2FDC = 2,5-furandicarboxylate(2-) ion. ip = isophthalate(2-) ion.

**Table 5 molecules-28-06263-t005:** Data for the thermogravimetric analysis of compound (**1**).

Step or R	Temp. (°C)	Time (min)	Weight Loss (%) Exp. Calc.		Evolved Gases or Residue (R)
1	30–115	0–12	6.905	8.098	~6 H_2_O *, CO_2_ (t *)
2	115–185	12–17	3.681	4.049	~3 H_2_O, CO_2_ (t *)
(1 + 2)	(30–185)	(0–17)	(10.586)	11.035	9 H_2_O
3	185–255	17–23	31.177	-	CO_2_, H_2_O
4	255–335	23–32	7.128	-	CO_2_, CO, H_2_O
5	335–450	32–45	28.922	-	CO_2_, CO, H_2_O, N_2_O_,_ NO, NO_2_, CH_4_ (t *)
R	520	95	19.532	17.879	4 CuO

* t = trace amounts. R = solid residue.

## Data Availability

Data are contained within the article or Appendix A.

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
