# Peer review of "H(N3)dap (Hdap = 2,6-Diaminopurine) Recognition by Cu2(EGTA): Structure, Physical Properties, and Density Functional Theory Calculations of [Cu4(μ-EGTA)2(μ-H(N3)dap)2(H2O)2]·7H2O"

_molecules, 2023, doi:10.3390/molecules28176263_

Round 1
Reviewer 1 Report
Comments for molecules-2578657
In this paper entitled "H(N3)dap (Hdap = 2,6-diaminopurine) recognition by Cu2(EGTA): Structure, physical properties and DFT-calculations of [Cu4(μ-EGTA)2(μ-H(N3)dap)2(H2O)2]·7H2O ", authors have reported the synthesis of a novel tetranuclear Cu(II) complex molecule having two μ-EGTA chelators as binding ligands and two 2,6-diaminopurine molecules as the tautomer H(N3)dap and then have studied it by single crystal X-ray diffractometry, various physical methods (thermal stability, spectral and magnetic properties), as well as DFT theoretical calculations.
Generally in this manuscript, some sentences are written very long and vague, which is not pleasant for the reader and must be revised. For example, pay attention to the first sentence of the abstract and introduction! Also, there are some writing problems in some parts.
According to the mentioned cases, I have some recommendation for the authors in order to improve the level of this article.
In title
● you have mentioned what word "Hdap" stands for, while in the abstract and in the introduction, you have mentioned this again, so there is no need to repeat it. In my opinion, remove this explanation from the title and only mention it in the abstract and use only the abbreviation word in your manuscript because the reader is familiar with your explanation. Please use this case for other abbreviations as well.
In Abstract
●In the third line after the name of the complex, you have used (1), But in different parts of his manuscript, the name of the compound is repeated along with the number considered. You can use only the specified number or symbol to avoid name confusion.
●What do you mean by "uncoordinated water is disordered" in the fifth line? Please explain more.
●In line 13,"in 1" should be change to "in Figure 1".
●You have used "Supramolecular Chemistry" in the keyword. It would be much better if you mentioned this word in the abstract.
Introduction
●In this section, you have mentioned that the reason for your research on the formation of a complex with "aminopurine" is because of the great attention that this compound has attracted at the moment. The question arises in the mind of the reader, why this compound has attracted so much attention? Please be sure to add the necessary explanations about this.
●Your introduction is very short and concise and needs more elaboration. You can provide more complete explanations about the other ligands mentioned and the reason for using them. Also, the reason for the synthesis of this complex is not mentioned at all, and the necessity of its use and efficiency is not specified. Please review again.
●In order to improve your introduction, I suggest that you read the following articles and refer to these references by adding a paragraph in this section.
https://doi.org/10.1016/j.ccr.2014.03.012
https://doi.org/10.1016/j.ccr.2015.10.004
https://doi.org/10.1039/C7DT00894E
https://doi.org/10.1021/cg501752e
https://doi.org/10.1039/C3DT51971F
In Results and discussion
●In section 2.1, line 5," (mw 221.1)" should be change to "( 221.1 g/mol)".
●In section 2.1, line 13, what do you mean by "unreacted apple green residue"? Can you explain more? This sentence is very vague.
●In section 2.3, you have mentioned that the IR spectrum of the binary compound and new compound 1, is given in Figure S1, but there is no trace of this figure in your article. Please check it.
●In the explanation section of Figure 4, why are the words "TGA,TSC and curves " bolded? Can you explain the need to use this distinction?
In Materials and methods
●In section 3.1, line 2, what do you mean by "in the absence of presence"? The reason for your use of two opposite words in this section is completely incomprehensible.
●In section 3.1, line 4," (mw 575.47)" should be change to "(575.47 g/mol)".
●In section 3.1, you stated that “The binary chelate has been prepared in good yields by the strategy previously reported from our group according to reference 16”, this subject is not mentioned at all in this reference. How do you justify this non-uniformity?
In reference
●In this section, references 1 and 2 are given, while in your entire manuscript, these two references are not referenced at all. Please review it more carefully.
●Reference [12] format is not compatible with other references, please add the following doi to the end of the reference:
https://doi.org/10.1107/S0108270186094647
need to be improved
Author Response
First, we would like to thank this referee for his careful reading of the manuscript, corrections and suggestions. We have revised the manuscript accordingly. Or replies are listed below:
- you have mentioned what word "Hdap" stands for, while in the abstract and in the introduction, you have mentioned this again, so there is no need to repeat it. In my opinion, remove this explanation from the title and only mention it in the abstract and use only the abbreviation word in your manuscript because the reader is familiar with your explanation. Please use this case for other abbreviations as well.
Reply: Done, thanks.
- In the third line after the name of the complex, you have used (1), But in different parts of his manuscript, the name of the compound is repeated along with the number considered. You can use only the specified number or symbol to avoid name confusion.
Reply: Done, thanks
- What do you mean by "uncoordinated water is disordered" in the fifth line? Please explain more.
Reply: We means that the lattice water molecules are disordered. This has been explained
- In line 13,"in 1" should be change to "in Figure 1".
Reply: Fixed, thanks
- You have used "Supramolecular Chemistry" in the keyword. It would be much better if you mentioned this word in the abstract.
Reply: We have adjusted the keywords
- In this section, you have mentioned that the reason for your research on the formation of a complex with "aminopurine" is because of the great attention that this compound has attracted at the moment. The question arises in the mind of the reader, why this compound has attracted so much attention? Please be sure to add the necessary explanations about this.
Reply: Thank you for this suggestion, we have improved the introduction.
- Your introduction is very short and concise and needs more elaboration. You can provide more complete explanations about the other ligands mentioned and the reason for using them. Also, the reason for the synthesis of this complex is not mentioned at all, and the necessity of its use and efficiency is not specified. Please review again.
Reply: We have re-written the introduction
- In order to improve your introduction, I suggest that you read the following articles and refer to these references by adding a paragraph in this section.
https://doi.org/10.1016/j.ccr.2014.03.012
https://doi.org/10.1016/j.ccr.2015.10.004
https://doi.org/10.1039/C7DT00894E
https://doi.org/10.1021/cg501752e
https://doi.org/10.1039/C3DT51971F
Reply: Thank you for this suggestion, we have incorporated these references at the end of the introduction
- In section 2.1, line 5," (mw 221.1)" should be change to "( 221.1 g/mol)".
Reply: Done, many thanks
- In section 2.1, line 13, what do you mean by "unreacted apple green residue"? Can you explain more? This sentence is very vague.
Reply: This sentence has been re-written
- In section 2.3, you have mentioned that the IR spectrum of the binary compound and new compound 1, is given in Figure S1, but there is no trace of this figure in your article. Please check it.
Reply: Figure S1 is in the ESI file
- In the explanation section of Figure 4, why are the words "TGA,TSC and curves " bolded? Can you explain the need to use this distinction?
Reply: The use of “bold” format has been eliminated
- In section 3.1, line 2, what do you mean by "in the absence of presence"? The reason for your use of two opposite words in this section is completely incomprehensible.
Reply: Fixed, many thanks
- In section 3.1, line 4," (mw 575.47)" should be change to "(575.47 g/mol)".
Reply: Fixed, thanks
- In section 3.1, you stated that “The binary chelate has been prepared in good yields by the strategy previously reported from our group according to reference 16”, this subject is not mentioned at all in this reference. How do you justify this non-uniformity?
Reply: The synthesis is indeed in reference 16 (it is provided as a reference, see reference 15 of that manuscript)
- In this section, references 1 and 2 are given, while in your entire manuscript, these two references are not referenced at all. Please review it more carefully.
Reply: Fixed, thanks
- Reference [12] format is not compatible with other references, please add the following doi to the end of the reference: https://doi.org/10.1107/S0108270186094647
Reply: Done, thanks
Reviewer 2 Report
This paper present data on new four-nuclear copper(II) complex with derivatives of adenine and EDTA. Ctystal data, results of other physic-chemical methods and calculations are in agreement.
But some comments:
1) In Introduction: Scheme 1 shows H(N9)dap, but in MS the authors describe H(N3)dap. It is desirable to give here the numbers of atoms, since the basicity order of N atoms is given in the text.
2) There is no atomic numbering scheme, so it is difficult to compare the types of H-bonds. For example, in the heading of Table 3 "…includes interactions of the N3-H···O and N6-H···O…" type, but the Table does not contain atoms with such numbers or even a separation of bonds into these two types.
3) If the authors discuss pi-pi-interactions, they must give some of their geometric characteristics.
4) Description of the structure as " zig-zag chains based on pi-pi-interactions" is not complete. Since there are quite strong intermolecular H-bonding O(27)…O(65)#1 and O(27)…O(22)#1 with the distance Cu(4)…Cu(4)#1 6.766 Å and Cu(4)…Cu(3)#1 6.791 Å. These H-bonds joint chains into layer.
And the main question: The use of the term "molecular recognition" to the described structure is something unexpected. The presence of stacking between molecules is not a sign of recognition.
Editing of English language required.
Very strange phrase: ''both organics determines the tetranuclear dimensionality of the complex " and so on.
Author Response
First, we would like to thank this referee for his careful reading of the manuscript, corrections and suggestions. We have revised the manuscript accordingly. Or replies are listed below:
This paper present data on new four-nuclear copper(II) complex with derivatives of adenine and EDTA. Ctystal data, results of other physic-chemical methods and calculations are in agreement.
But some comments:
1) In Introduction: Scheme 1 shows H(N9)dap, but in MS the authors describe H(N3)dap. It is desirable to give here the numbers of atoms, since the basicity order of N atoms is given in the text.
Reply: Figure 1 has been modified as requested.
2) There is no atomic numbering scheme, so it is difficult to compare the types of H-bonds. For example, in the heading of Table 3 "…includes interactions of the N3-H···O and N6-H···O…" type, but the Table does not contain atoms with such numbers or even a separation of bonds into these two types.
Reply: The atom numbering scheme is given in Figure 2b
3) If the authors discuss pi-pi-interactions, they must give some of their geometric characteristics.
Reply: Done, see Figure 3.
4) Description of the structure as " zig-zag chains based on pi-pi-interactions" is not complete. Since there are quite strong intermolecular H-bonding O(27)…O(65)#1 and O(27)…O(22)#1 with the distance Cu(4)…Cu(4)#1 6.766 Å and Cu(4)…Cu(3)#1 6.791 Å. These H-bonds joint chains into layer.
Reply: Thank you for this suggestion. These H-bonds have been described in addition to the π-stacking interactions. A new Figure 3b has been generated.
And the main question: The use of the term "molecular recognition" to the described structure is something unexpected. The presence of stacking between molecules is not a sign of recognition.
Reply: We agree with the referee. This has been corrected.
Reviewer 3 Report
1. Some typesetting errors could be revised, such as “alternating π, π-interactions…” should be revised as “alternating π ̶ π interactions.
2. it is only “new”, NOT “novel”, I did not suggest the authors use such word.
3. In the synthesized section, I cannot found the detail quantity of each starting material
4. Please give the axis direction for each figure.
5. Please provide the EA data and PXRD for complex, it should be checked its purity before do the magnetism properties.
6. Some work on the magnetism and DFT work could be referred, such as Mol. Struct. 1291(2023)136074, Theor. Chem. Acc. 2022, 141, 68; ACS Omega, 2018, 3, 17986−17990; Monatsh Chem, 2017, 48,1259–1267.
7. In the magnetism analyze, pls compared the related document from the magnetic exchange.
work
Author Response
First, we would like to thank this referee for his careful reading of the manuscript, corrections and suggestions. We have revised the manuscript accordingly. Or replies are listed below:
- Some typesetting errors could be revised, such as “alternating π, π-interactions…” should be revised as “alternating π ̶ π interactions.
Reply: The manuscript has been revised to correct the typos and grammar
- it is only “new”, NOT “novel”, I did not suggest the authors use such word.
Reply: Changed as requested
- In the synthesized section, I cannot found the detail quantity of each starting material
Reply: This is provided in section 3.1
- Please give the axis direction for each figure.
Reply: Done
- Please provide the EA data and PXRD for complex, it should be checked its purity before do the magnetism properties.
Reply: The EA has been already provided at the end of section 3.1 Moreover a picture of the crystals is given in Figure S3 that prove the purity of the compound. Unfortunately, we cannot generate the PXRD for complex 1 because the instrument in broken at this moment. However, to our opinion the purity of the sample has been already proven.
- Some work on the magnetism and DFT work could be referred, such as Mol. Struct. 1291(2023)136074, Theor. Chem. Acc. 2022, 141, 68; ACS Omega, 2018, 3, 17986−17990; Monatsh Chem, 2017, 48,1259–1267.
Reply: We have cited and commented the first manuscript related to magnetism. However, the second, third and fourth theoretical manuscripts are totally unrelated (rotation of methyl groups, fullerene investigation, halogen bonding respectively)
- In the magnetism analyze, pls compared the related document from the magnetic exchange.
Reply: Done, see end of section
Round 2
Reviewer 1 Report
I recommend acceptance of the present version of the manuscript
Need improvement
Reviewer 2 Report
1) Lines 225-235: change "degrees Celsius" for "°C"
2) Line 270: "cm3K/mol" should be "cm3K/mol".
I understand that the calculations take a lot of time, but it is desirable to estimate the energy of the intermolecular H-bonds described on page 5 (lines 164-167). According to Mercury (CSD materials/Calculations/UNI Intermolecular potentials), it is comparable to those given for staking interactions.
Reviewer 3 Report
accept